# Reliability and Validity of Laboratory and Field Cardiorespiratory Exercise Tests for Wheelchair Users: A Systematic Review

**DOI:** 10.3390/ijerph22030384

**Published:** 2025-03-06

**Authors:** Iker Garate, Javier Yanci, Josu Ascondo, Aitor Iturricastillo, Cristina Granados

**Affiliations:** 1Physical Education and Sport Department, Faculty of Education and Sport, University of the Basque Country, UPV/EHU, 01007 Vitoria-Gasteiz, Spain; iker.garate@ehu.eus; 2Society, Sports and Physical Exercise Research Group (GIKAFIT), Physical Education and Sport Department, Faculty of Education and Sport, University of the Basque Country, UPV/EHU, 01007 Vitoria-Gasteiz, Spain; javier.yanci@ehu.eus (J.Y.); cristina.granados@ehu.eus (C.G.); 3AKTIBOki: Research Group in Physical Activity, Physical Exercise and Sport, Physical Education and Sport Department, Faculty of Education and Sport, University of the Basque Country, UPV/EHU, 01007 Vitoria-Gasteiz, Spain; josu.ascondo@ehu.eus

**Keywords:** cardiopulmonary exercise test, disability, physiology, endurance, physical activity

## Abstract

Background: cardiorespiratory fitness is one of the most important components of physical fitness. In this paper, we set out to identify cardiopulmonary tests evaluated for measurement properties in wheelchair users and determine which are reliable and valid for this population. Methods: Articles were collected from PubMed, Scopus, SPORTDiscus, and Web of Science. The initial search was conducted in October 2022 and updated in July 2023 for recent publications. From 1257 screened studies, 42 met the criteria: (a) participants were wheelchair users, (b) tests measured cardiorespiratory fitness, (c) test reliability or validity was reported, (d) articles were original, and (e) full text was in English. Two independent researchers extracted participant details (number, gender, age, disability) and test information, with a third researcher resolving disagreements. Statistical analyses of test reliability and validity were documented. Results: Methodological quality was assessed using the COSMIN checklist. Evidence levels for reliability and validity were established. Moderate evidence was found for reliability in one field test, and for validity in two lab and two field tests. Conclusions: While most tests show good reliability and validity, sample sizes are small, limiting conclusions. Further research is needed to strengthen the evidence and identify the most suitable tests for wheelchair users. Researchers are encouraged to replicate validation studies to support future testing.

## 1. Introduction

The wheelchair-using population comprises a wide range of people with disabilities, such as tetraplegia, paraplegia, spina bifida, cerebral palsy, sclerosis or amputations [1], Nevertheless, physical activity is highly recommended for all of them in order to improve their physical fitness and also their quality of life [1,2,3,4]. One of the most important components of physical fitness is cardiorespiratory fitness (CRF), a significant factor in morbidity and mortality [5,6,7]. CRF can be measured directly, expressed as maximal oxygen consumption (VO_2_max), or estimated from the peak work rate achieved on a treadmill. Adequate cardiorespiratory fitness may improve the quality of life of wheelchair users, because it is associated with better health [6], reduced mortality [5,6], greater functional capacity and greater autonomy to move around without depending on others [8]. Due to different factors, such as impairments, activity limitations or participation restrictions, people with disabilities are more likely to be physically inactive [9,10,11], have poorer cardiorespiratory fitness, and tend to develop more chronic diseases and comorbidities [9,12,13]. For this reason, measuring cardiorespiratory fitness in wheelchair users is crucial.

Cardiorespiratory fitness is commonly measured in a laboratory test [14,15,16]. These tests are performed in controlled and stable situations, and they typically measure variables such as gas exchange [16,17,18], blood lactate concentration [16,17,18,19,20,21] or power output [22,23,24]. However, as laboratory tests require expensive and sophisticated equipment, qualified personnel and a high investment of time, field tests have also been used for wheelchair users to measure or estimate cardiorespiratory fitness, because they allow a large number of participants to be tested cheaply, in less time and with greater ecological validity [25,26]. As in the general population, both laboratory [27] and field [28] protocols have been used for wheelchair users. Due to differences in this population’s functional and situational capacities, the protocols used to measure cardiorespiratory fitness in people with disabilities have often been adapted from protocols used for the general population [29]. For example, Eriksson et al. [30] adapted the traditional maximal protocols used in able-bodied people for wheelchair users by creating a fixed resistance roller and measuring wheel speed. Later, more sophisticated systems were utilized for the same purpose. Klaesner et al. [31] developed a dynamometer, adjustable to any wheelchair, that could simulate different resistances and slopes and measure the forces. Arm-crank ergometers [24] or treadmills with special harnesses that attach to wheelchairs to stabilize the person in them [32] have also been used. In the same way, the field protocols used in the ordinary population have also been adapted to the characteristics and needs of wheelchair users [33]. For example, Vanderthommen et al. [34] adapted Leger and Boucher’s shuttle run test [35] for wheelchair tennis players and Yanci et al. [36] did the same with the Yo-Yo recovery test in wheelchair basketball.

Regardless of whether they are laboratory or field tests, when any variable is measured, the tool used (in this case a cardiopulmonary test) must be valid and reliable [37]. This means that it has to measure what it is designed to measure, and whenever it measures the same thing, it must give the same result [37]. The availability of valid and reliable laboratory and field protocols for wheelchair users allows their cardiorespiratory capacity to be measured adequately. This is important for identifying people who could benefit from a prevention program [38] or even for sporting purposes [33]. This allows wheelchair users to be classified by level, and in the case of large groups, to be sorted into subgroups for individualized programs. Whatever the objective of the exercise program, testing is necessary to prescribe, monitor and evaluate it [22,39] correctly. For this reason, it seems essential to deepen out scientific knowledge on the validity and reliability of laboratory and field tests for wheelchair users.

Most of the protocols used in the general population have been widely validated; however, they suffer from important variations in adjusting to the needs/characteristics of wheelchair users [36,40,41,42,43]. Therefore, the validity and reliability of these new/adapted protocols should be analyzed. Nevertheless, although there are studies on the validity of cardiopulmonary tests for wheelchair users, the conclusions that can be drawn are limited due to the reduced number of participants [22,44,45]. For this reason, this study aims to unify the information obtained in those studies, assess their quality, and show the existing scientific evidence for using each test. Similar reviews have been performed in other populations or with other types of tests as targets [29,46,47,48,49], but not for cardiorespiratory tests for wheelchair users. Eerden et al. [27] gathered the existing cardiopulmonary laboratory tests for wheelchair users with spinal cord injury, but did not assess their reliability and/or validity. The only review that is close to our topic is the one by Gossey-Tolfrey et al. [28], but this is not a systematic review, so it only includes a few articles on this topic, and it does not assess the quality of the studies; it only focuses on field tests, and the validity and reliability of the tests is not the main research topic. Therefore, given that there is no comparable systematic review and the topic to be addressed is relevant, the main aim of this review is to analyze the existing scientific evidence on the validity and reliability of laboratory and field tests for measuring cardiorespiratory fitness in wheelchair users.

## 2. Materials and Methods

### 2.1. Search Strategy

This systematic review was conducted using the PRISMA (Preferred Reporting Items for Systematic Reviews and Meta-Analysis) recommendations [50]. Research articles were gathered using the PubMed, Scopus, SPORTDiscus, and Web of Science database platforms, representing databases from multiple health and physical activity disciplines. The literature search was conducted in October 2022 and repeated in July 2023 for recently published articles. The search strategy included the following keywords with the relevant Boolean operators inserted: (“disability” OR “physical impairment” OR “wheelchair” OR “cerebral palsy” OR “amputation”) AND (“cardio*” OR “aerobic capacity” OR “respiratory”) AND “test” AND (“reliability” OR “validity”). The protocol was registered and published in OSF (https://doi.org/10.17605/OSF.IO/G45BR, accessed on 1 February 2025).

### 2.2. Eligibility Criteria

To be included in this systematic review, studies needed to meet the following criteria: (a) at least 50% of participants had to be regular wheelchair users, either for everyday life or sporting purposes; (b) the test under study had to measure or estimate cardiorespiratory fitness; (c) the article had to provide information on reliability or validity of the test; (d) studies had to be original articles published in peer-reviewed impact journals; and (e) full-text articles had to be written in English. No publication date limit was set. Failure to meet any of the criteria meant being excluded from the review. No cardiovascular cycling tests were included in this systematic review.

### 2.3. Screening

The selection process is presented in Figure 1. The search results were merged, and duplicate records of the same document were removed. This resulted in 1257 articles. Identified articles on the systematic search were initially checked for relevance by 2 independent researchers (the first and last authors). Articles were selected after a sequential reading of the title and abstract, always in this order. Subsequently, the researchers reviewed the full texts of potentially eligible articles. A third researcher (the second author) resolved reviewer disagreements regarding the study’s inclusion. The references of the articles that were fully read were consulted to identify possible additional studies. In the case of articles found in systematic reviews, they were considered for inclusion only when the full text was available.

### 2.4. Data Extraction

Data extraction was performed by 2 independent researchers (the first and last authors) and supported by a third researcher (the second author) when necessary (Figure 1). The authors critically analyzed the selected articles and extracted the data on participants (number, gender, age, disability, and characteristics), the tests studied and statistical analysis on reliability and validity.

### 2.5. Quality Assessment

The methodological quality of the articles was evaluated using the COSMIN (Consensus-Based Standards for Selecting Health Measurement Instruments) checklist [51]. This methodology was used in previous reviews to assess the quality of studies on the measurement properties of different tests in other populations [29,47,48]. The checklist consists of 10 boxes that evaluate different measurement properties as “inadequate”, “doubtful”, “adequate” and “very good”. For this review, Boxes 6 (relative reliability), 7 (absolute reliability), 8 (criterion validity) and 9 (convergent validity) were rated if applicable. The overall score for each box is determined by the lowest score obtained on any of the items [29].

### 2.6. Level of Evidence

Only studies with “doubtful” or better methodological quality were used for the evidence level analysis [29]. Evidence was considered “strong” if there was 1 study with “very good” quality or multiple studies with “adequate” quality with similar results; “moderate” if there was 1 study with “adequate” quality or multiple studies with “doubtful” quality with similar results; and “limited” if there was only 1 study with “doubtful” quality [29].

## 3. Results

### 3.1. Quality of Included Studies

A total of 42 studies were included in the review (Table 1): 16 assessed both the reliability and validity of a test, 13 assessed only reliability and 13 assessed only the validity; therefore, 29 studies assessed reliability and 29 assessed validity. The quality of reliability studies was rated as “inadequate” 24 times, “doubtful” 4 times and “adequate” once. The quality of validity studies was rated as “inadequate” 11 times, “doubtful” 14 times and “adequate” 3 times.

### 3.2. Characteristics of the Participants

A total of 42 studies were included in the review. Regarding gender, 22 studies were mixed, 19 included only men, and 1 article was composed entirely of women. The total number of participants was 1065 (236 females and 829 males). The studies’ sample sizes varied from 6 persons [18,20] to 102 persons [57], and the age range was 4–83 years. A total of 6 articles were focused on young people (4–18 years) [43,61,64,66,67,71], 1 article studied young and adult people together (14–46 years) [56], and the participants in the remaining 35 articles were adults (>18 years). Only three studies included people older than 60 years in their samples [17,69,73]. Concerning the disabilities of the participants, the samples in 15 papers were composed of people with spinal cord injuries (2 of them included people with poliomyelitis too), 5 articles were focused on people with cerebral palsy, 2 studies’ participants had unilateral amputations and 1 article studied people with osteogenesis imperfecta. The remaining 19 articles had people with different disabilities in the sample. In 18 studies, only wheelchair athletes were included. 

### 3.3. Laboratory Tests

Twenty of the included studies evaluated laboratory tests. The total number of laboratory tests studied was 29. Thirteen protocols were performed in a wheelchair ergometer: eleven were incremental maximal tests (six by increasing the resistance, four by increasing the speed and one by increasing the cadence), and two were simulated races. A further six protocols were carried out with an arm-crank ergometer, of which four were incremental maximal tests, and two were submaximal tests. Another seven protocols were conducted by pushing a wheelchair on a treadmill, including six incremental maximal tests (two by increasing inclination and speed, three by increasing inclination only and one by increasing speed only) and one submaximal test. The last three protocols were incremental maximal tests performed on an arm–crank ergometer or recumbent stepper. All laboratory protocols are summarized in Table 2.

Among the 20 articles, 7 assessed both the reliability and validity of a test, 9 assessed only the reliability, and 4 assessed only the validity. A total of 16 reliability reports and 11 validity reports were registered. The quality of reliability reports was evaluated as “inadequate” in 13 cases and “doubtful” in 3 cases. The quality of validity reports was “inadequate” in six cases, “doubtful” in three cases and “adequate” in two cases. There were no “very good” reports. Considering the best quality articles, two tests showed a moderate evidence level for validity: the maximal wheelchair ergometer resistance test [23] and the 6-min arm test [17]. Both tests had limited evidence for reliability. No laboratory test showed moderate or strong evidence for reliability (Table 3).

### 3.4. Field Tests

Twenty-two of the included studies evaluated field tests. The total number of field tests studied was 18. A total of 12 protocols were maximal incremental wheelchair tests (8 performed in a straight line, 2 in a “figure of eight”, 1 in an octagon and 1 on a 400 m track), of which 3 were intermittent tests, and the rest were continuous. Another five tests involved covering a distance as long as possible over a limited time (5, 6 or 12 min), but in two of them, the pushing cadence was marked with a metronome. In the other four, the participants were free to perform as best they could. The remaining test was a submaximal incremental test. All the field protocols are summarized in Table 4.

Among the 22 articles, 9 assessed both the reliability and validity of a test, 4 assessed only the reliability and 9 assessed only the validity. A total of 13 reliability reports and 18 validity reports were registered. The quality of reliability reports was evaluated as “inadequate” in 11 cases, “doubtful” in 1 case and “adequate” in 1 case. The quality of validity reports was “inadequate” in 5 cases, “doubtful” in 11 cases and “adequate” in 2 cases. There were no “very good” reports. Considering the best quality articles, the shuttle wheelchair test [67] and the adapted Léger and Boucher test [58] showed a moderate evidence level for validity. Only one test showed a moderate evidence level for reliability: the 6-min push test [43] (Table 5).

## 4. Discussion

The main aim of this review was to analyze the existing scientific evidence on the reliability and validity of laboratory and field tests for measuring cardiorespiratory fitness in wheelchair users. To the authors’ knowledge, this is the first systematic review to analyze the level of evidence of measurement properties of cardiopulmonary tests for wheelchair users. The validity and reliability of cardiorespiratory fitness assessments are of paramount importance when conducting laboratory and field tests. High validity ensures that measures reflect true fitness levels, while reliability guarantees consistent results across different testing conditions, enhancing the effectiveness of fitness evaluations [74]. The results showed that although several studies had measured the reliability and validity of cardiopulmonary tests for wheelchair users, there was no test with strong evidence. The main reason for this was the small sample size of the studies, which is very common when studying a particular population, because it is often difficult to gather many participants. Therefore, different studies should analyze the same tests, instead of trying to evaluate new ones, as most existing studies have positive results. However, according to Janssen et al. [75], there were too many tests (57 different tests in 42 articles), and a need for unification between them.

As is often the case in general physical activity studies [76], with only 75 women (22.2%), women had less presence than men (77.8%) in the included studies. Most of the studies were mixed, with 19 involving only men and 1 involving only women. The sample sizes varied widely, ranging between 6 and 102 persons. Regarding age, most of the investigations were conducted in adults; some studies focused on young people, and very few focused on people older than 60 years. Regarding disability type, most studies were carried out with samples of individuals with different impairments or people with spinal cord injuries; there were few studies on other disabilities or impairments, such as cerebral palsy, amputations or osteogenesis imperfecta. Another important thing to consider is that 16 studies were conducted with an entirely athletic population. It is likely that, in some cases, tests with good reliability and validity values in a concrete sample may not be suitable for other samples [24]. Moreover, given that they are a population that can benefit significantly in terms of health from understanding aerobic fitness [1,2,3,4], more studies on validity and reliability in wheelchair users are needed, especially for women, in people under 18 and over 60 years of age and for some concrete disabilities.

In relation to the laboratory tests, many different tests were analyzed (29 tests) in the included studies (21 studies). Thus, the only test studied in 2 different articles was the maximal wheelchair ergometer speed test [20,21]. Therefore, although most of the studies have shown positive results, due to the limited samples and lack of corroboration, there is currently a low level of evidence to determine whether one laboratory test is more appropriate. The highest level of evidence was obtained for the maximal wheelchair ergometer resistance test [17] and 6-min arm test [23], with both obtaining a moderate level of evidence for validity. However, both had limited evidence for reliability. This shows the need for further studies on the reliability and validity of laboratory tests for wheelchair users. Furthermore, it is unclear which type of laboratory test is the gold standard for this population, so more research should be carried out on this issue. Thus, Bloemen et al. [64] recommended using a wheelchair ergometer over an arm ergometer because they obtained higher oxygen consumption and heart rate values. This is likely due to both the specificity of the movement and the involvement of more muscle mass. The results of Hartung et al. [44] suggested that when performing a wheelchair treadmill laboratory test, combining increases in both incline and resistance was more appropriate, because higher peak values were obtained by increasing both of them than by increasing the incline or resistance alone (in the resistance protocol, they obtained the lowest peak values). Other studies [69,77] suggested that for certain individuals, such as unilateral amputees, it may be more appropriate to use a recumbent stepper, as this would involve more muscle mass, obtaining higher peak values, as in the aforementioned cases. These results highlight the possible need to use different laboratory tests depending on the type of population and measurement objectives. Also, there may not be a single gold-standard test.

Regarding the field tests, the three most analyzed tests appeared in three articles each: the shuttle wheelchair test [61,66,67], the adapted Léger and Boucher test [54,55,58] and the multistage octagonal field test [34,63,65]. All three tests were continuous maximal multistage wheelchair tests: the first over a 10 m straight, the second on a 400 m athletics track, and the third in a 15 × 15 m octagon. The 6-min push test (performed over a 10-m straight) was analyzed in two articles, and the remaining tests appeared in only one. Most field tests analyzed were maximal multistage wheelchair tests (12 tests), which increased speed until exhaustion. These tests can be intermittent or continuous, and can be performed on different routes (e.g., going around an athletics track, going back and forth in a straight line, turning around an octagon or performing a figure eight-shaped route). Having rests between bouts or changes in direction can affect the outcome of the test. De Groot et al. [33] found that the test they used with tennis players was more related to skills (such as turning and the ability to propel the chair correctly at high speeds) than cardiorespiratory fitness. They argued that the increase in speed is the primary limiting factor of the test, and that an increase in resistance, often seen in laboratory tests, is the most appropriate way to obtain maximal oxygen consumption values. This is in line with the aforementioned results of Hartung et al. [44], who obtained worse values in the laboratory test when they increased only the speed. The remaining six tests included a submaximal field test [52] and five time-limited tests, two of which required participants to maintain a constant cadence [39], whereas the other three had no cadence limitations [40,41,43,71]. Different routes were also used in this type of test, such as 200 m indoor tracks, basketball courts or 10 and 15 m straights, which could affect the result. Only two time-limited tests obtained positive results: the 6-min push tests, with one over a 10 m straight [43,71] and the other over a 15 m straight [40]. This might suggest that 6 min is more appropriate than 12 min for this type of test, and that limiting the cadence may not be the most appropriate. This agrees with Christensen et al. [22], who report that upper limb tests should last between 5 and 9 min, unlike lower limb tests. Regarding evidence levels, there were two field tests with moderate evidence for validity: the shuttle wheelchair test [67] and the adapted Léger and Boucher test [58]. On the other hand, there was only one field test with a moderate level of evidence for reliability: the 6 min push test over a 10 m straight [43]. Therefore, although good results were obtained in most of the articles, there is not even moderate evidence to consider any field test as having good reliability and validity for wheelchair users, so future studies should study the psychometric properties of the above-mentioned tests for this population.

### Study Limitations

Although this review was conducted with high methodological and scientific rigor standards, it is not without limitations. The literature search was conducted in English, so studies in other languages will have been left out of this review; four databases were used for this purpose, and, undoubtedly, some studies will have been missed. Furthermore, the limitation that at least 50% of participants had to be wheelchair users also meant that interesting studies with non-disabled people were not included in this review.

## 5. Conclusions

The main conclusion of this review is that most studies appear to obtain good levels of reliability and validity, but are sparse and lack large samples to draw solid conclusions, so more studies evaluating existing tests are needed to strengthen the level of evidence. Furthermore, many similar tests, but different protocol variations, make it impossible to group the results and draw more consistent conclusions. This is why we encourage researchers to replicate studies that assess the validity and reliability of cardiopulmonary tests in wheelchair users in order to establish which laboratory and field tests are most appropriate for this population. In particular, there is a lack of studies on validity and reliability for wheelchair users in women, children, adolescents, people over 60 years of age and people with some specific disabilities.

## Figures and Tables

**Figure 1 ijerph-22-00384-f001:**
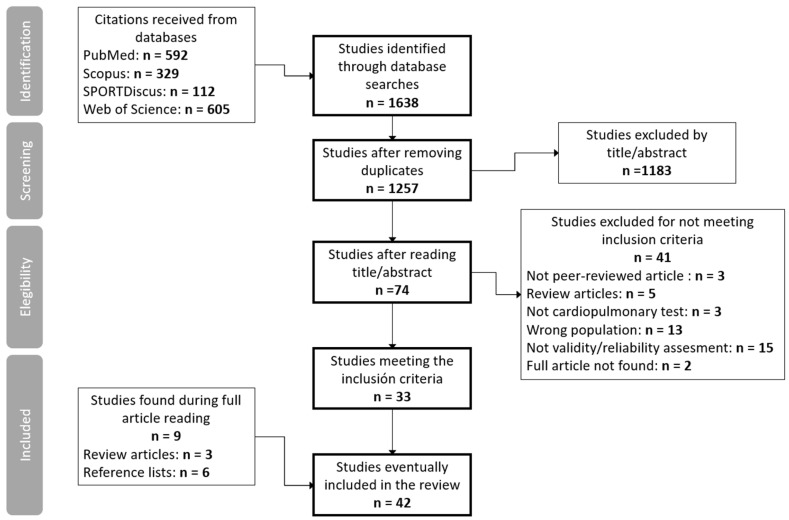
Study selection flow-chart.

**Table 1 ijerph-22-00384-t001:** Evaluation of the included studies according to the COSMIN checklist.

Study	OR	OV	R1	R2	R3	R4	R5	R6	R7	R8	R9	V1	V2	V3	V4	V5	V6 *	V7 *
Wicks et al. [16]	-	4										3	2	1	4	2	2	1
Kofsky et al. [52]	-	3										2	2	3	1	1		
Bhambhani et.al. [19]	4	-	1	1	2	1	1	4	-	3	1							
Langbein et al. [17]	3	2	1	1	2	1	3	2	-	3	1	2	1	2	1	1		
Bhambhani et al. [20]	4	3	1	1	2	1	2	4	-	3	1	3	3	1	1	1		
Bhambhani et al. [21]	4	-	1	1	2	1	2	4	-	3	1							
Hartung et al. [44]	4	4	1	1	2	3	2	4	-	2	1	3	2	1	4	2	2	1
Bhambhani et al. [18]	-	3										1	3	2	1	1		
Holland et al. [53]	4	-	1	1	2	1	4	4	-	3	1							
Longimuir et al. [42]	3	4	1	1	1	1	1	3	-	3	1	2	2	4	4	1		
Vinet et al. [54]	-	3										2	3	3	1	1		
Dwyer et al. [41]	-	3										1	3	3	1	3		
Poulain et al. [55]	4	-	1	1	2	1	1	4	1	4	1							
Vanlandewijck et al. [56]	4	3	1	1	2	1	1	4	-	3	1	2	3	2	1	1		
Stewart et al. [57]	4	-	1	1	3	4	2	1	-	2	1							
Vinet et al. [58]	-	2										2	2	1	1	1		
Vanderthomen et al. [34]	4	3	1	1	1	1	1	4	1	2	1	2	3	1	1	1		
Laskin et al. [39]	4	3	1	1	2	1	2	4	1	1	1	2	3	2	1	1		
Hol et al. [23]	3	2	1	1	2	1	2	3	2	1	1	2	2	2	1	1		
Goosey-Tolfrey et al. [59]	4	3	1	1	2	1	1	4	1	1	1	1	3	2	1	1		
Leicht et al. [60]	4	-	1	1	1	1	1	4	2	2	1							
Cowan et al. [40]	3	4	1	1	2	1	2	3	1	1	1	1	2	2	4	3		
Verschuren et al. [61]	4	3	1	1	2	1	1	4	1	1	1	2	3	2	1	1		
Verschuren et al. [43]	2	-	1	1	2	1	1	2	1	1	1							
Goosey-Tolfrey et al. [62]	-	4										1	3	2	4	1		
Weissland et al. [63]	-	4										1	1	1	4	2	2	1
Bloemen et al. [64]	4	4	1	1	1	1	1	4	1	1	1	3	3	3	4	1		
Yanci et al. [36]	4	-	1	1	1	1	1	4	-	2	1							
Weissland et al. [65]	-	4										3	1	3	4	2	2	1
de Groot et al. [33]	-	3										2	3	1	1	1		
Bongers et al. [66]	4	3	1	1	2	1	1	4	1	1	1	2	3	1	1	1		
Gauthier et al. [4]	4	-	1	1	1	1	1	4	1	1	1							
Bloemen et al. [67]	4	2	1	1	2	1	1	4	1	1	1	1	2	2	1	1		
Kelly et al. [45]	4	-	1	1	1	1	1	4	2	2	1							
Pérez Tejero et al. [68]	-	3										2	3	2	1	1		
Simmelink et al. [69]	4	-	1	1	2	1	3	4	1	1	1							
Christensen et al. [22]	4	-	1	1	2	1	2	4	1	1	1							
Morgan et al. [24]	4	3	1	1	2	1	1	4	1	1	1	2	3	2	1	1		
Qi et al. [70]	-	4										1	3	1	4	1		
Damen et al. [71]	4	4	1	1	2	1	1	4	1	1	1	1	1	2	4	1		
Goosey-Tolfrey et al. [72]	-	4										2	3	2	4	1		
Holm et al. [73]	4	-	1	1	1	1	1	4	-	1	1							

Abbreviations: 1 = very good; 2 = adequate; 3 = doubtful; 4 = inadequate; OR = overall reliability score; OV = overall validity score; R1 = uses at least two measurements; R2 = ensures that the administrations will be independent; R3 = ensures that the patients will be stable in the interim period on the construct to be measured, R4 = uses an appropriate time interval between the two measurements, which is long enough to prevent recall, and short enough to ensure that patients remain stable; R5 = ensures that the test conditions will be similar for the measurements (e.g., type of administration, environment, instructions); R6 = performs the analysis in a sample with an appropriate number of patients (taking into account expected number of missing values); R7 = for continuous scores: calculate the standard error of measurement, smallest detectable change or limits of agreement; R8 = for continuous scores: calculates an intraclass correlation coefficient (ICC); R9 = provides a clear description of how missing items will be handled; V1 = describes whether the proposed criterion can be considered as a reasonable ‘gold standard’/formulate hypotheses about expected relationships between the PROM under study and other outcome measurement instrument(s); V2 = performs the analysis in a sample with an appropriate number of patients (taking into account expected number of missing values)/provides a clear description of the construct(s) measured by the comparator instrument(s); V3 = uses an appropriate time schedule for assessments of the PROM of interest and ‘gold standard’/Use comparator instrument(s) with sufficient measurement properties; V4 = for continuous scores: calculates correlations, or the area under the receiver operating curve/performs the analysis in a sample with an appropriate number of patients (taking into account expected number of missing values); V5 = provides a clear description of how missing items will be handled/used an appropriate time schedule for assessments of the PROM of interest and comparison instruments; V6 = uses statistical methods that are appropriate for the hypotheses to be tested; V7 = provides a clear description of how missing items will be handled. * if columns V6 and V7 are filled in, it means that convergent validity was evaluated, so the items evaluated in columns V1–V5 are those indicated in second position.

**Table 2 ijerph-22-00384-t002:** Characteristics and reliability and validity results of laboratory tests.

Study	Test	Tests Characteristics	Disability/Sport	Participants Sex and Age (Years)	Reliability (Relative and/or Absolute)	Criterion (CRI) or Convergent (CON) Validity
Wicks et al. [16]	Maximal WE speed test	20 rpm + 10 or 5 rpm/min with low resistance	SCI (n = 2M were able-bodied)	n = 3F 4M 28.1 ± 4.1 (23–34)		CON No significant differences
	Maximal WE speed test	20 rpm + 10 or 5 rpm/min with high resistance				
	Maximal ACE resistance test	60 rpm + 25-50-100 kpm/min				
Bhambhani et al. [19]	Maximal WE cadence test	40 strikes/min + 8 strikes/min/2 min	PP and TT	n = 2F 5M 26.5 ± 3.5 (22–32)	r = 0.98 * *t* test no differences	
Langbein et al. [17]	Maximal WE resistance test	6 W + 5-20 W/3 min (continuous and intermittent)	TT, PP, amputations and lower limb fractures	n = 51M (17–69)	r = 0.9 *	CRI No significant differences r = 0.91 *
Bhambhani et al. [20]	Maximal WE speed test	5 km/h + 2 km/h/2 min	CP athletes, national and international level	n = 6M 24.8 ± 3.7 (19–29)	r = 0.89 *	CRI r < 0.31 (n = 4)
Bhambhani et al. [21]	Maximal WE speed test	5 km/h + 2 km/h/2 min with 1 min rest	CP athletes, national and international level	n = 11M 25.1 ± 4.3 (19–33)	r = 0.89 *, 0.9 *	
Hartung et al. [44]	Maximal TM grade and speed test	3.2 km/h + 1.6 km/h & 0.5% every 2 min	PP and low extremity polio disability	n = 7M 33 (21–44)	ICC = 0.86 *, 0.87 *	CON ANOVA, no differences
	Maximal TM grade test	0 + 1%/2 min at 4.8 km/h			ICC = 0.41 *, 0.44 *	
	Maximal TM speed test	3.2 + 3.2 km/h/2 min			ICC = 0.40 *, 0.83 *	
Bhambhani et al. [18]	Simulated 1.6 km race in WE	1.6 km simulated race	TT Actives	n = 7M 30.6 ± 5.2		CRI r = 0.79 *, 0.81 *, 0.23, 0.32
	Simulated 3.2 km race in WE	3.2 km simulated race	PP Actives	n = 6M 29.0 ± 4.6		CRI r = 0.82 *, 0.88 *, 0.19, 0.61
Holland et al. [53]	Maximal WE speed test	4 km/h + 1 km/h/2 min	CP	n = 1F 4M 25.6 ± 3.7 (21–31)	r = 0.79 *, 0.83 * (Data mixed with cycle ergometer test)	
Longimuir et al. [42]	Arm CAFT	Submaximal 3 × 3 min in ACE (Stopping when achieving a certain HR, load dependant on sex and age)	Different physical disabilities	n = 18F 17 M (21–59)	r = 0.97 * (n = 30)	CRI Paired *t* test, no differences
Stewart et al. [57]	Maximal ACE resistance test/maximal WE resistance test	15 W + 15 W/3 min +7.5 W/3 min (after RPE15) with 2 min recovery	SCI	n = 19F 83M 29.7 ± 10.3	ICC = 0.82 *	
Hol et al. [23]	6-MAT	Submaximal 6 min at 60–70% HR max in ACE	PP and TT	n = 5F 25M 36.3 ± 9.3 (19–49)	ICC = 0.81 * (0.58–0.92) SEM = 1.62 mL/kg/min	CRI r = 0.92 *, 0.73 *, 0.63 *
Leicht et al. [60]	Maximal TM grade test	Individualized speed 1% + 0.3/min or 0.1/40 s	WR, WB and WT; national and international	n = 24M 28.1 ± 5.2 (TT) 31.7 ± 8.7 (PP) 24.0 ± 6.2 (non-SCI)	ICC = 0.84–0.99 * CV = 9.3% (TT), 4.5% (PP), 3.3% (non-SCI)	
Goosey-Tolfrey et al. [62]	Submaximal TM speed test	6 × 4 min	WR and WB players; elite level	n = 26M 30 ± 5 (TT) 29 ± 9 (PP) 27 ± 8 (non-SCI)		CRI TE = 0.14–0.16 L/min (TT) 0.19–0.22 L/min (PP) 0.54–68 L/min (non-SCI) CV = 10.1–11.1% (TT) 6.5–7.5% (PP) 16.8–20.2% (non-SCI)
Bloemen et al. [64]	Maximal WE resistance test	0 + 0.1 torque/min at 60–80 rpm	Spina bifida wheelchair users	n = 4F 9M 13.4 ± 3.5 (8–17)		CRI t test, no difference VO_2_peak 15% greater than ACE test
				n = 11F 13M 14.8 ± 3.0 (8–19)	ICC = 0.93 * (0.83–0.97) SEM = 1.87 mL/kg/min SDC = 5.18 mL/kg/min	
Gauthier et al. [4]	Maximal TM grade and speed test	Different slope and speed increases every 1 min	Wheelchair users	n = 21F 4M 35.3 ± 14.9 (18–63)	ICC = 0.84 * (0.66–0.92), 0.86 * (0.70–0.93) SDC = 2.27 mL/kg/min, 157.9 mL/min SEM = 5.3 mL/kg/min, 368.5 mL/min	
Simmelink et al. [69]	Maximal ALE resistance test	3 min 20 W + 10 W/min at 50 rpm	Unilateral lower limb amputation	n = 3F 14M 54.5 ± 18.6 (25–80)	ICC = 0.84 * (0.61–0.94), 0.91 * (0.77–0.97) LoA = −0.56 to 0.60 L/min	
Christensen et al. [22]	Maximal ACE resistance test	40 W+ 20 W/min	Unilateral lower limb amputation	n = 8M 32.5 ± 4.6 (18–40)	ICC = 0.51 * (0.11–0.85), 0.73 * (0.40–0.93), 0.74 * (0.40–0.93) CV = 14.48% SEM = 0.18 L/min LoA = −0.66 to 0.40 L/min	
Morgan et al. [24]	Maximal WE resistance test	Increasing resistance every minute at 70% of max speed	SCI 70% athletes	n = 10M 33 ± 19.6 (18–60)	ICC = 0.82 *, 0.97 * LoA = −6 to 4.5 mL/kg/min	CRI r = 0.79 *, 0.77 * t test, no differences LoA = −4.1 to 3.6 mL/kg/min
Qi et al. [70]	Maximal WE resistance test	10 W + 5 or 10 W/min at 1 m/s	SCI	n = 3F 7M 42.1 ± 8.4 (29–55)		CRI ICC = 0.91 *
Holm et al. [73]	Maximal ACE resistance test	5 W + 10 W/min at 60 rpm	SCI	n = 1F 4M 47 ± 25.6 (21–83)	ICC = 0.99 *	
	mTBRS-XT	25 W + 15 W/2 min in ALE	SCI	n = 3F 6M 61.7 ± 13.3 (41–76)	ICC = 0.96 *	
	TBRS-XT	50 W + 25 W/2 min in ALE	SCI	n = 1F 7M 44.3 ± 15.6 (27–75)	ICC = 0.99 *	

* *p* < 0.05; abbreviations: 6-MAT = six-minute arm test; ACE = arm-crank ergometer; ALE = arm–leg ergometer; CAFT = Canadian aerobic fitness test; CP = cerebral palsy; CV = coefficient of variation; ICC = intraclass correlation coefficient; LoA = limits of agreement; mTBRS-XT = modified total body recumbent stepper exercise test; PP = paraplegia; r = correlation coefficient; SCI = spinal cord injury; SDC = smallest detectable change; SEM = standard error measurement; TBRS-XT = total body recumbent stepper exercise test; TM = treadmill; TT = tetraplegia; VO_2_ = oxygen consumption volume; WB = wheelchair basketball; WE = wheelchair ergometer; WR = wheelchair rugby; WT = wheelchair tennis.

**Table 3 ijerph-22-00384-t003:** Evidence level of reliability and validity of laboratory tests (with a limited or greater evidence level).

Studies	Test	Reliability	Validity
Langbein et al. [17]	Maximal WE resistance test	Limited	Moderate
Hol et al. [23]	6-MAT	Limited	Moderate
Bhambhani et al. [20]	Maximal WE speed test	?	Limited
Morgan et al. [24]	Maximal WE resistance test	?	Limited
Bhambhani et al. [18]	Simulated 1.6 km race in WE	N/A	Limited
Bhambhani et al. [18]	Simulated 3.2 km race in WE	N/A	Limited
Longimuir et al. [42]	Arm CAFT	Limited	?

Abbreviations: ? = inadequate methodology; 6-MAT = six-minute arm test; CAFT = canadian aerobic fitness test; N/A = not available; WE = wheelchair ergometer. Note: those tests that did not reach a level of evidence of at least limited reliability or validity are not presented in this table.

**Table 4 ijerph-22-00384-t004:** Characteristics and reliability and validity results of field tests.

Study	Test	Tests Characteristics	Disability/Sport	Participants Sex and Age (Average, Standard Deviation and Range)	Reliability (Relative and Absolute)	Criterion (CRI) or Convergent (CON) Validity
Kofsky et al. [52]	Submaximal field test	3 × 5 min at 40-60-80% HRmax with 2 min rest	Lower limb disabilities; sedentary individuals, active individuals and athletes	n = 7F 28.7 ± 3.4 (18–55)		CRI r = 0.61 *
				n = 42M 28.2 ± 1.3 (18–55)		CRI r = 0.67 *
Vinet et al. [54]	ALBT	4 km/h + 1 km/h/min in 400 m tartan	WT and WRA	n = 9M 28.9 ± 4.2 (24–35)		CRI r = 0.65 * Wilcoxon, no differences LoA ≃ −7 to 14 mL/kg/min
Dwyer et al. [41]	12WPT	Maximal distance in 12 min in 200 m indoor tartan	WB national level	n = 13F 26 ± 6 (19–40)		CRI r = 0.46 *, 0.30
Poulain et al. (1999) [55]	ALBT	4 km/h + 1 km/h/min in 400 m tartan	WT, WRA, WTT; regional, national and international level	n = 8M 30.8 ± 5.1 (24–39)	No significant differences (MANOVA) LoA ≃ −2 to 1.2 km/h CV = 2%	
Vanlandewijck et al. [56]	SWT	5 km/h + 0.5 km/h/min between two marks 25 m away	WB national level	n = 20M 31.7 ± 10.4 (14–46)	r = 0.97 *	CRI r = 0.64 *, 0.87 *, (n = 15)
Vinet et al. [58]	ALBT	4 km/h + 1 km/h/min in 400 m tartan	WB, WRA, WF or AS	n = 9F 40M 30.3 ± 0.4 (18–47)		CRI r = 0.81 *
Vanderthomen et al. [34]	MFT in octagon	6 km/h + 0.37 km/h/min in a 15 m × 15 m octagon	PP and post-polio	n = 10M 38.2 ± 13.0	ICC = 0.99 *, 0.88 * LoA ≃ −1.4 to 1.8 stages −4.4/4.6 mL/kg/min	CRI r = 0.77 *
Laskin et al. [39]	Cadence-based submaximal field test	5 min 60 pushes/min in a basketball court	WB, elite level	n = 24M 26.1 ± 6.6	ICC = 0.50 * LoA = −0.83/1.05 L/min (n = 16)	CRI r = 0.49 * t test, no differences
	Cadence-based submaximal field test	5 min 80 pushes/min in a basketball court	WB, elite level	n = 24M 26.1 ± 6.6	ICC = 0.62 * LoA = −0.66 to 1.06 L/min (n = 16)	CRI r = 0.56 * t test, no differences
Goosey-Tolfrey et al. [59]	SWT	8.5 km/h + 0.5 km/h/min between two marks 20 m away	WB, international level	n = 24M 29 ± 6	ICC = 0.88 * (0.58–0.97), 0.91 * (0.69–0.98) SEM = 2.4 beats/min, 86 m CV = 1.3%, 4.4% (n = 10)	CRI r = 0.57 *
Cowan et al. [40]	6MPT	Maximal distance in 6 min surrounding two cones 15 m away	PP and TT	n = 6F 34M 34 ± 10 (20–45)	ICC = 0.97 * (0.94–0.98) (All), 0.93 (0.80–0.98) (TT), 0.97 (0.93–0.99) (PP) LoA ≈ −70 to 65 m (All)	CRI ICC = 0.86 * (0.75–0.92) (All), 0.86 (0.70–0.93) (PP), 0.65 (0.17–0.87) (TT)
Verschuren et al. [61]	SWT	2 km/h + 0.25 km/h/min between two marks 10 m away	Spastic CP	n = 5F 18M 13.3 ± 3.6	ICC = 0.99 * (0.98–1.00) SEM = 0.5 min SDC = 1.4 min LoA ≃ −1.45 to 1.45 stages	CRI r = 0.84 * (n = 15)
Verschuren et al. [43]	6MPT	Maximal distance in 6 min surrounding two marks 10 m away	Spastic CP	n = 22F 51M 11.8 ± 3.6 (4–18)	ICC = 0.97 * (0.96–0.98) SEM = 20.9 m SDC = 57.9 m LoA ≃ −54.8 to 62.2 m	
Weissland et al. [63]	MFT in octagon	6 km/h + 0.37 km/h/min in a 15 m × 15 m octagon	WB, national level	n = 2F 14M 32.4 ±5.3 (23–41)		CON Student’s *t* test, differences r = 0.93 *, 0.84 *
	MFT in 8 figure	6 km/h + 0.37 km/h/min in 32 m × 15 m “8 figure”				
Yanci et al. [36]	Yoyo intermittent recovery test	2 × 10 m with 10 s active recovery increasing speed	WB, national level	n = 2F 14 M 33.1 ± 7.4 (21–46)	ICC = 0.94 *	
Weissland et al. [65]	MFT in octagon	6 km/h + 0.37 km/h/min in a 15 m × 15 m octagon	WB, national level	n = 2F 16M 32.0 ± 5.7 (22–41)		CON Student’s *t* test, no differences r = 0.84 * LoA ≃ −8 to 7 mL/kg/min
	30-15ITF	6 km/h + 0.5 km/h/45 s in 40 m (30 s push–15 s rest)				
de Groot et al. [33]	SWT	5 km/h + 0.32 km/h/min between two marks 20 m away	WT, national and international level	n = 15M 21.2 ± 8.4		CON r = 0.40, 0.47
Bongers et al. [66]	SWT	2 km/h + 0.25 km/h/min between two marks 10 m away	Osteogenesis imperfecta wheelchair users	n = 5F 8 M 15.5 ± 6.4 (9–25)	ICC = 0.95 * (0.83–0.98), 0.97 * (0.89–0.99), 0.92 * (0.71–0.98) SEM = 0.7 stages SDC = 1.9 stages LoA = −2.5 to 1.35 stages *t* test no differences	CRI r = 0.61 *, 0.45
Bloemen et al. [67]	SWT	2 km/h + 0.25 km/h/min between two marks 10 m away	Spina bifida wheelchair users, in daily life or in sports	n = 17F 16M 14.5 ± 3.1 (5–18)		CRI r = 0.85 *, 0.84 * t test, no differences
				n = 12F 16M 14.7 ± 3.3 (5–18)	ICC = 0.96 * (0.92–0.98), 0.93 * (0.84–0.97), 0.98 * (0.96–0.99) CV = 6.2%, 6.4% SDC = 1.5 shuttles SEM = 0.5 shuttles	
Kelly et al. [45]	30-15IFT-28m	8 km/h + 0.5 km/h/min in 28 m (30 s push–15 s rest)	WR, international level	n = 10M 31.8 ± 7.3 (20–44)	ICC = 0.99 * SEM = 1.02 km/h CV = 1.9% LoA = −0.51 to 0.61 km/h	
Pérez Tejero et al. [68]	SWT	6 km/h + 0.5 km/h/min between two marks at 28 m	WB, elite level	n = 7M 33.3 ± 6.0 (27–44)		CRI r = 0.85 *
Damen et al. [71]	6MPT	Maximal distance in 6 min surrounding two cones 10 m away	Spina bifida wheelchair users, in daily life or in sport	n = 21F 32M (n = 8 lost) 13.6 ± 3.8 (5–19)		CRI 86% (18.5) of maximal VO_2_
				n = 14F 12M (n = 5 lost) 13.6 ± 3.8 (5–19)	ICC = 0.95 * (0.83–0.98) SDC = 60.7 m SEM = 21.9 m	
Goosey-Tolfrey et al. [72]	MFT in 8 figure	6.5 km/h + 0.36 km/h/min in 28.2 m × 10.8 m “8 figure”	WR, national level	n = 16M 28 ± 6		CRI Paired *t* test, no differences

* *p* < 0.05; abbreviations: 6MPT = six-minute push test; 12WPT = twelve-minute wheelchair push test; ALBT = adapted Léger and Boucher test; AS = adapted swimming; CP = cerebral palsy; CV = coefficient of variation; HR = heart rate; ICC = intraclass correlation coefficient; ITF = intermittent fitness tests; LoA = limits of agreement; MFT = multistage field test; PP = paraplegia; r = correlation coefficient; SDC = smallest detectable change; SEM = standard error measurement; SWT = shuttle wheelchair test; TT = tetraplegia; VO_2_ = oxygen consumption volume; WB = wheelchair basketball; WF = wheelchair fencing; WR = wheelchair rugby; WRA = wheelchair racing; WT = wheelchair tennis; WTT = wheelchair table tennis.

**Table 5 ijerph-22-00384-t005:** Evidence level of reliability and validity of field tests (with a limited or greater evidence level).

Studies	Test	Reliability	Validity
Bloemen et al. [67]; Bongers et al. [66]; Verschuren et al. [61]	SWT	?	Moderate
Vinet et al. [54]; Vinet et al. [58]	ALBT	N/A	Moderate
Verschuren et al. [43]	6MPT	Moderate	?
Vanlandewijck et al. [56]	SWT	?	Limited
Vanderthomen et al. [34]	MFT in octagon	?	Limited
Laskin et al. [39]	Cadence-based submaximal field test	?	Limited
Goosey-Tolfrey et al. [59]	SWT	?	Limited
Kofsky et al. [52]	Submaximal field test	N/A	Limited
Dwyer et al. [41]	12WPT	N/A	Limited
de Groot et al. [33]	SWT	N/A	Limited
Pérez Tejero et al. [68]	SWT	N/A	Limited
Cowan et al. [40]	6MPT	Limited	?

Abbreviations: ? = inadequate methodology; 6MPT = six-minute push test; 12WPT = twelve-minute wheelchair push test; ALBT = adapted Léger and Boucher test; MFT = multistage field test; N/A = not available; SWT = shuttle wheelchair test. Note: those tests that did not reach a level of evidence of at least limited reliability or validity are not presented in this table.

## Data Availability

No new data were created.

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
