# Peer review of "Reliability and Validity of Laboratory and Field Cardiorespiratory Exercise Tests for Wheelchair Users: A Systematic Review"

_ijerph, 2025, doi:10.3390/ijerph22030384_

Round 1

Reviewer 1 Report

Comments and Suggestions for Authors

 you should read about cycling and its cardiovascular tests.

You talk about cardiorespiratory fitness but do not mention the variables/dimensions/categories that compose it. Your introduction will be much better if you add these points.

Materials and methods

Please clarify whether the review was registered in PROSPERO or another platform.

Results

Although I understand what has been done, I cannot understand the order of the articles in the tables. Is it by year? Alphabetically? Level reached?

Discussion

Dear authors, this section is undoubtedly very important, and due to the quality of your work, you should make some changes to enhance it.

In many paragraphs, you do not have citations for the statements made. On the other hand, you should reorganize this section. One suggestion is:

1. Problematic: Highlight the importance of solving this problem by citing some articles that discuss the importance of validity and reliability.

2. Separate the discussion into laboratory tests and field tests, where each should have: i) Critical analysis of your main results, ii) How your results converse with other reviews on wheelchair users, iii) group limitations that the articles have and mention them.

Conclusions

The authors should review their conclusions since, apparently, more than analyzing, you are evaluating the degree of validity and reliability. In addition, you should clarify what type of validity you are carrying out: construct validity or content validity.

Author Response

Comments and Suggestions for Authors

Authors’ response (AR): We sincerely thank the reviewer for the valuable advice and suggestions on this manuscript. We greatly appreciate the positive feedback and recommendations for improvement. Your constructive criticism has been especially helpful. We have addressed your concerns point by point, and the revisions made to the manuscript are highlighted in blue colour.

You should read about cycling and its cardiovascular tests.

AR: Thank you for the suggestion. Our study took into account the arm crank ergometer, arm-leg ergometer, treadmill, and field test; unfortunately, cycling tests were not included. So, to clarify, we have added this sentence “No cardiovascular cycling tests were included in this systematic review.” in Eligibility criteria to avoid misunderstandings. Nevertheless, thank you for your suggestion.

You talk about cardiorespiratory fitness but do not mention the variables/dimensions/categories that compose it. Your introduction will be much better if you add these points.

AR: You are right. That is why we added some cardiorespiratory fitness variables in the Introduction.

“One of the most important components of physical fitness is cardiorespiratory fitness (CRF), a significant factor in morbidity and mortality [5-7]. CRF can be measured directly, expressed as maximal oxygen consumption (VO2max), or estimated from the peak work rate achieved on a treadmill. Adequate cardiorespiratory fitness may improve the quality of life of wheelchair users because it is associated with better health [6], reduced mortality [5,6], greater functional capacity and greater autonomy to move around without depending on others [8].

Format:

Materials and methods

Please clarify whether the review was registered in PROSPERO or another platform.

AR: The systematic review was registered in OSF, so the following sentence has been added: “The protocol was registered and published in OSF (https://doi.org/10.17605/OSF.IO/G45BR).”

Results

Although I understand what has been done, I cannot understand the order of the articles in the tables. Is it by year? Alphabetically? Level reached?

AR: Thank you for the comment. The order of the articles in the tables is by year, from the oldest to the most current.

Discussion

Dear authors, this section is undoubtedly very important, and due to the quality of your work, you should make some changes to enhance it.

AR: Thank you for your suggestion. We have made some changes to the discussion to enhance it.

In many paragraphs, you do not have citations for the statements made. On the other hand, you should reorganize this section. One suggestion is:

  1. Problematic: Highlight the importance of solving this problem by citing some articles that discuss the importance of validity and reliability.

AR: Thank you for the suggestion. We have added those sentences in the discussion:

“The main aim of this review was to analyze the existing scientific evidence on the reliability and validity of laboratory and field tests for measuring cardiorespiratory fitness in wheelchair users. To the authors’ knowledge, this is the first systematic re-view to analyze the level of evidence of measurement properties of cardiopulmonary tests for wheelchair users. The validity and reliability of cardiorespiratory fitness assessments are of paramount importance when conducting laboratory and field tests. High validity ensures that measures reflect true fitness levels, while reliability guarantees consistent results across different testing conditions, enhancing the effectiveness of fitness evaluations [74]. The results showed that although several studies had measured the reliability and validity of cardiopulmonary tests for wheel-chair users, there was no test with strong evidence.”.

  1. Shushan T, Lovell R, McLaren SJ, Buchheit M, Dello Iacono A, Arguedas-Soley A, Norris D. Assessing criterion and longitudinal validity of submaximal heart rate indices as measures of cardiorespiratory fitness: A preliminary study in football. J Sci Med Sport. 2024, 27(8), 565-571. doi: 10.1016/j.jsams.2024.04.006.

  1. Separate the discussion into laboratory tests and field tests, where each should have: i) Critical analysis of your main results, ii) How your results converse with other reviews on wheelchair users, iii) group limitations that the articles have and mention them.

AR: Thank you for the idea. The discussion is separated into laboratory and field tests (third and fourth paragraphs of the discussion). Concerning the first point, we understand that the critical analysis of your main results has been carried out. Moreover, we believe that the second point is not the aim of our study. In relation to the third point, study limitations are added at the end of the discussion.

Thank you very much!

Conclusions

The authors should review their conclusions since, apparently, more than analyzing, you are evaluating the degree of validity and reliability. In addition, you should clarify what type of validity you are carrying out: construct validity or content validity.

AR: Thank you for the suggestion. The main aim of this review is to analyze the existing scientific evidence on the validity and reliability of laboratory and field tests for measuring cardiorespiratory fitness in wheelchair users. That is why we give more importance to evaluating the degree of validity and reliability. On the other hand, the document analyzes criterion or convergent validity.

Reviewer 2 Report

Comments and Suggestions for Authors

Psychometric Properties of Cardiorespiratory Exercise Tests

In the abstract, "psychometric properties" describes "cardiorespiratory exercise tests." However, psychometric properties typically refer to measurements related to psychological constructs, such as reliability and validity in psychological assessments. It would be more accurate to refer to the tests' "measurement properties" or "physiological validity." Please clarify or revise this terminology to better align with the context.

Further Research and Timeframe Considerations

The statement "Further research is needed to..." is noted, but discussing whether selecting a different timeframe might have influenced the outcomes could be beneficial. Could variations in data collection periods, such as seasonal impacts or longer durations, provide additional insights? Expanding on this aspect would strengthen the study's conclusions.

Significance of the Study

The importance of the study has not been clearly articulated. It would be helpful to emphasize why this research is necessary and how it contributes to exercise testing for wheelchair users. Providing a clear justification will enhance the manuscript's impact.

Rationale for Timeframe Selection

The chosen timeframe, from October 2022 to July 2023, is not explained. What criteria were considered when selecting this period? Were there practical, seasonal, or methodological reasons for this choice? Addressing this question would provide better transparency and contextual understanding of the study design.

The manuscript will improve clarity, relevance, and scientific rigour by addressing these points.

Comments on the Quality of English Language

Minor editing. 

Author Response

Comments and Suggestions for Authors

Authors’ response (AR): We sincerely thank the reviewer for the valuable advice and suggestions on this manuscript. We greatly appreciate the positive feedback and recommendations for improvement. Your constructive criticism has been especially helpful. We have addressed your concerns point by point, and the revisions made to the manuscript are highlighted in blue colour.

Psychometric Properties of Cardiorespiratory Exercise Tests

In the abstract, "psychometric properties" describes "cardiorespiratory exercise tests." However, psychometric properties typically refer to measurements related to psychological constructs, such as reliability and validity in psychological assessments. It would be more accurate to refer to the tests' "measurement properties" or "physiological validity." Please clarify or revise this terminology to better align with the context.

AR: Thank you for your suggestion. You are right; we have changed the “measurement properties” in the abstract.

Further Research and Timeframe Considerations

The statement "Further research is needed to..." is noted, but discussing whether selecting a different timeframe might have influenced the outcomes could be beneficial. Could variations in data collection periods, such as seasonal impacts or longer durations, provide additional insights? Expanding on this aspect would strengthen the study's conclusions.

AR: Due to the limited extent of words in the abstract, we have written like this, but at the end of the document, in the conclusions, we give more ideas to provide additional insights.

Significance of the Study

The importance of the study has not been clearly articulated. It would be helpful to emphasize why this research is necessary and how it contributes to exercise testing for wheelchair users. Providing a clear justification will enhance the manuscript's impact.

AR: Thank you for the comment. The study's significance is stated in the last paragraph of the introduction, but following your suggestion, we have added this sentence to the discussion.

“To the authors’ knowledge, this is the first systematic review to analyze the level of evidence of measurement properties of cardiopulmonary tests for wheelchair users. The validity and reliability of cardiorespiratory fitness assessments are of paramount importance when conducting laboratory and field tests. High validity ensures that measures reflect true fitness levels, while reliability guarantees consistent results across different testing conditions, enhancing the effectiveness of fitness evaluations [74]. The results showed that although several studies had measured the reliability and validity of cardiopulmonary tests for wheelchair users, there was no test with strong evidence.”.

  1. Shushan T, Lovell R, McLaren SJ, Buchheit M, Dello Iacono A, Arguedas-Soley A, Norris D. Assessing criterion and longitudinal validity of submaximal heart rate indices as measures of cardiorespiratory fitness: A preliminary study in football. J Sci Med Sport. 2024, 27(8), 565-571. doi: 10.1016/j.jsams.2024.04.006.

Rationale for Timeframe Selection

The chosen timeframe, from October 2022 to July 2023, is not explained. What criteria were considered when selecting this period? Were there practical, seasonal, or methodological reasons for this choice? Addressing this question would provide better transparency and contextual understanding of the study design.

The manuscript will improve clarity, relevance, and scientific rigour by addressing these points.

AR: Thak you for the suggestion. The data of the search strategy was from October 2022 to July 2023 because that was the period in which we started and finished the article search. We had to write to some authors to obtain their articles in PDF, and that’s why there was a long period.

Reviewer 3 Report

Comments and Suggestions for Authors

Dear authors,

Congratulations on the quality of the paper.

The systematic review aimed to identify laboratory and field cardiorespiratory exercise tests for wheelchair users, conducted in 4 databases, in accordance to the recommendations of PRISMA and using the COSMIN checklist to assess the methodological quality.

This topic is relevant in order to find evidence for reliability and validity for field and lab tests for wheelchair users. Overall, this review underscores the need for identifying the most suitable tests for wheelchair users. Therefore, in my opinion the paper is suitable for publication in the International Journal of Environmental Research and Public Health.

A specific comment is address to the abstract:

L. 19 –“Background” – this sentence is not suitable for this topic, in my perspective this is the aim.

Author Response

Comments and Suggestions for Authors

Dear authors,

Congratulations on the quality of the paper.

The systematic review aimed to identify laboratory and field cardiorespiratory exercise tests for wheelchair users, conducted in 4 databases, in accordance to the recommendations of PRISMA and using the COSMIN checklist to assess the methodological quality.

This topic is relevant in order to find evidence for reliability and validity for field and lab tests for wheelchair users. Overall, this review underscores the need for identifying the most suitable tests for wheelchair users. Therefore, in my opinion the paper is suitable for publication in the International Journal of Environmental Research and Public Health.

Authors’ response (AR): We sincerely thank the reviewer for the valuable advice and suggestions on this manuscript. We greatly appreciate the positive feedback and recommendations for improvement. Your constructive criticism has been especially helpful. We have addressed your concerns point by point, and the revisions made to the manuscript are highlighted in blue colour.

A specific comment is address to the abstract:

  1. 19 –“Background” – this sentence is not suitable for this topic, in my perspective this is the aim.

AR: Thank you for this suggestion. We have added a sentence in the background:

“Background: One of the most important components of physical fitness is cardiorespiratory fitness. Identify cardiopulmonary tests evaluated for measurement properties in wheelchair users and determine which are reliable and valid for this population.”

Reviewer 4 Report

Comments and Suggestions for Authors

Dear Authors,

The topic and findings are interesting and certainly needed. However, I have comments that may affect the reliability of this manuscript.

The introduction contains blank lines, as well as a discussion, that should be removed. It is also too long. The purpose needs to be redefined. "Existing scientific evidence on the validity" is unnecessary.

The article is also written clearly and legibly. Congratulations.

Reviewer

Author Response

Comments and Suggestions for Authors

Dear Authors,

The topic and findings are interesting and certainly needed. However, I have comments that may affect the reliability of this manuscript.

The introduction contains blank lines, as well as a discussion, that should be removed. It is also too long. The purpose needs to be redefined. "Existing scientific evidence on the validity" is unnecessary.

The article is also written clearly and legibly. Congratulations.

Reviewer

Authors’ response (AR): We sincerely thank the reviewer for the valuable advice and suggestions on this manuscript. We greatly appreciate the positive feedback and recommendations for improvement. Your constructive criticism has been especially helpful. We have made some changes in the introduction and in the discussion sections to enhance it.

Round 2

Reviewer 1 Report

Comments and Suggestions for Authors

Dear authors

After reading your article, I confirm that you have responded to or modified all the suggestions raised in the first review, on my part.

Reviewer 2 Report

Comments and Suggestions for Authors

Dear author(s), I congratulate you on the revisions.